# Generative networks for precision enthusiasts

Anja Butter[1], Theo Heimel[1], Sander Hummerich[1], Tobias Krebs[1],
Tilman Plehn[1], Armand Rousselot[2] and Sophia Vent[1]

**1** Institut für Theoretische Physik, Universität Heidelberg, Germany
**2** Heidelberg Collaboratory for Image Processing, Universität Heidelberg, Germany

## Abstract

Generative networks are opening new avenues in fast event generation for the LHC. We show how generative flow networks can reach percent-level precision for kinematic distributions, how they can be trained jointly with a discriminator, and how this discriminator improves the generation. Our joint training relies on a novel coupling of the two networks which does not require a Nash equilibrium. We then estimate the generation uncertainties through a Bayesian network setup and through conditional data augmentation, while the discriminator ensures that there are no systematic inconsistencies compared to the training data.

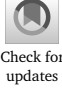

## Content

# 1   Introduction

Precise first-principle simulations provided by the theory community are a defining feature of
Large Hadron Collider (LHC) physics. They are based on perturbative quantum field theory
with fundamental Lagrangians as their physics input, and they provide the simulated events
necessary for modern LHC analyses. Because of the close correlation of complexity and preci-
sion in perturbative calculations, precision and speed are, largely, two sides of the same medal.
Both of these sides are facing major challenges for the LHC Runs 3 and 4, and the hope is that
machine learning and its modern numerics toolbox allow us to provide the simulations needed
for a 25-fold increase of LHC data as compared to Run 2.

In recent years, modern machine learning has shown great potential to improve LHC sim-
ulations. Underlying techniques include generative adversarial networks (GANs) [1–3], vari-
ational autoencoders (VAEs) [4, 5], normalizing flows [6–10], and their invertible network
(INN) variant [11–13]. As part of the standard LHC event generation chain [14], modern
neural networks can be applied to the full range of phase space integration [15, 16], phase
space sampling [17–20], amplitude computations [21, 22], event subtraction [23], event un-
weighting [24, 25], parton showering [26–30], or super-resolution enhancement [31, 32].
Conceptionally new developments are, for instance, based on fully NN-based event genera-
tors [33–37] or detector simulations [38–48]. In essence, there is no aspect of the standard
event generation chain that cannot be improved through modern machine learning.

A structural advantage of generative networks for event generation or detector simula-
tions is that, unlike forward Monte Carlo simulations, the network-based generation can be
inverted. Specifically, conditional GANs and INNs allow us to invert the simulation chain to
unfold detector effects [49, 50] and extract the hard scattering process at parton level in a
statistically consistent manner [51]. Because of their superior statistical properties, the same
conditional INNs can be used for simulation-based inference based on high-dimensional and
low-level data [52]. Finally, normalizing-flow or INN generators provide new opportunities
when we combine them with Bayesian network concepts [53–58] to construct uncertainty-
controlled generative networks [59].

In this paper we combine the full range of ML-concepts to build an NN-based LHC event
generator which meets the requirements in terms of phase space coverage, precision, and
control of different uncertainties. We first present a precision INN generator in Sec. 2 which
learns underlying phase space densities such that kinematic distributions are reproduced at
the percent level, consistent with the statistical limitations of the training data. Next, our
inspiration by GANs leads us to construct the DiscFlow discriminator–generator architecture
to control the consistency of training data and generative network in in Sec. 3. Finally, in Sec. 4
we illustrate three ways to control the network training and estimate remaining uncertainties
(i) through a Bayesian generative network, (ii) using conditional augmentations for systematic
or theory uncertainties, and (iii) using the DiscFlow discriminator for controlled reweighting.
While we employ forward event generation to illustrate these different concepts, our results
can be directly transferred to inverted simulation, unfolding, or inference problems.

# 2   Precision generator

As we will show in this paper, generative networks using normalizing flows have significant
advantages over other network architectures, including GANs, when it comes to LHC event
generation. As a starting point, we show how flow-based invertible networks can be trained
to generate events and reproduce phase space densities with high precision. Our network
architecture accounts for the complication of a variable number of particles in the final state.

## 2.1 Data set

The kind of NN-generators we discuss in this paper are trained on unweighted events at the hadronization level. We exclude detector effects because they soften sharp phase space features, so simulations without them tend to be more challenging and their results are more interesting from a technical perspective. This means our method will work even better on reconstucted objects.

The production of leptonically decaying $Z$-bosons with a variable number of jets is an especially challenging benchmark process. First, the network has to learn an extremely sharp $Z$-resonance peak. Second, QCD forces us to apply a geometric separation between jets, inducing a non-trivial topology of phase space. Finally, again because of QCD it does not make sense to define final states with a fixed number of jets, so our generative network has to cover a final state with a variable number of dimensions. Given these considerations we work with the process

$$pp \to Z_{\mu\mu} + \{1, 2, 3\} \text{ jets} \,, \tag{1}$$

simulated with SHERPA2.2.10 [60] at 13 TeV. We use CKKW merging [61] to generate a merged sample with up to three hard jets including ISR, parton shower, and hadronization, but no pile-up. The final state of the training sample is defined by FASTJET3.3.4 [62] in terms of anti-$k_T$ jets [63] with

$$p_{T,j} > 20 \text{ GeV}\,, \qquad \text{and} \qquad \Delta R_{jj} > R_{\min} = 0.4 \,. \tag{2}$$

The jets and muons are ordered in $p_T$. Because jets have a finite invariant mass, our final state dimensionality is three for each muon plus four degrees of freedom per jet, giving us phase space dimensionalities 10, 14, and 18. Momentum conservation does not further reduce the dimensionality, as not every generated hadron is captured by the three leading jets. However, we will reduce this dimensionality by one by removing the symmetry on the choice of global azimuthal angle. Our combined sample size is 5.4M events, divided into 4.0M one-jet events, 1.1M two-jet events, and 300k three-jet events. This different training statistics will be discussed in more detail in Sec. 4.1.

To define a representation which makes it easier for an INN to learn the kinematic patterns we apply a standard pre-processing. First, each lepton or reconstructed jet is represented by

$$\{ p_T, \eta, \phi, m \} \,. \tag{3}$$

Because we can extract a global threshold in the jet $p_T$ we represent the events in terms of the variable $\tilde{p}_T = \log(p_T - p_{T,\min})$. This form leads to an approximately Gaussian distribution, matching the Gaussian latent-space distribution of the INN. Second, the choice of the global azimuthal angle is a symmetry of LHC events, so we instead train on azimuthal angles relative to the muon with larger transverse momentum in the range $\Delta\phi \in [-\pi, \pi]$. A transformation into $\widetilde{\Delta\phi} = \text{atanh}(\Delta\phi/\pi)$ again leads to an approximately Gaussian distribution. For all phase space variables $q$ we apply a centralization and normalization step

$$\tilde{q}_i = \frac{q_i - \overline{q_i}}{\sigma(q_i)} \,. \tag{4}$$

Finally, we apply a whitening/PCA transformation separately for each jet multiplicity.

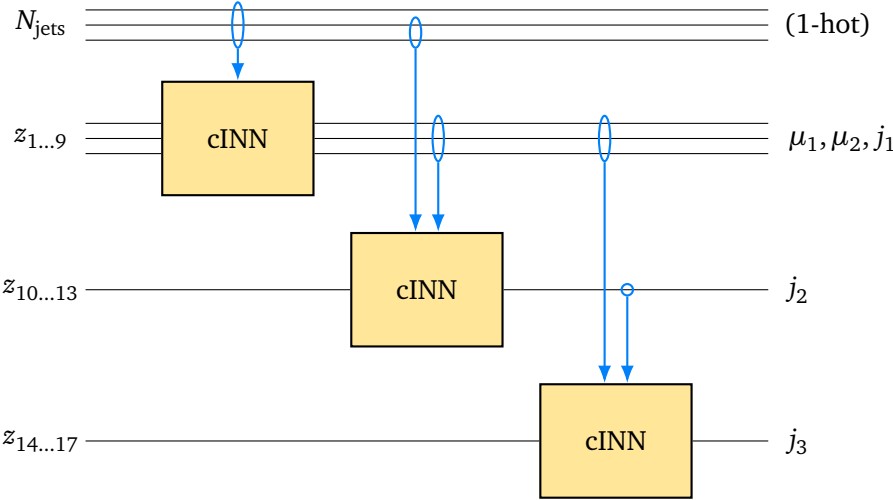

Figure 1: Generative flow architecture for events with two muons and one to three jets. The INNs relate the latent space (left) to the physical phase space (right).

## 2.2 INN generator

For a fixed final-state dimensionality we can use a completely standard INN [11,59] to generate LHC events, especially after the preprocessing step defined above. Our technical challenge is to expand the INN architecture to generate final states with 9, 13, and 17 phase space dimensions. Of course, we could just split the training sample into different multiplicities and train a set of individual networks. However, in this case each of these networks has to learn the basic QCD patterns, making this naive approach inefficient and unstable.

To increase the efficiency of the training, we use one network for the common $\mu_{1,2}$ and $j_1$ momenta and add additional small networks for each additional jet, as illustrated in Fig. 1. Some basic kinematic features of the muons and the first jet, like their transverse momentum balance, depend on possible additional jets, so we first provide the base network with the one-hot encoded number of jets as a condition. This allows the base network to generate all relevant $\{\mu\mu j\}$-configurations. Starting from those configurations we then train additional networks for each additional jet. These small networks are conditioned on the training ob-

Table 1: Training setup and hyperparameters for the INN generators used in our different setups.

| hyperparameter | INN (Sec. 2.2) | INN (Sec. 3.1) | DiscFlow (Sec. 3.2) | BINN (Sec. 4.1) |
|---|---|---|---|---|
| LR scheduling | one-cycle | same | same | same |
| Starter LR | $10^{-4}$ | $4 \cdot 10^{-4}$ | $2 \cdot 10^{-4}$ | $10^{-5}$ |
| Maximum LR | $10^{-3}$ | $4 \cdot 10^{-3}$ | $2 \cdot 10^{-3}$ | $10^{-4}$ |
| Epochs | 100 | 200 | 200 | 100 |
| Batch size | 1024 | 2048 | 2048 | 3072 |
| ADAM $\beta_1, \beta_2$ | 0.9, 0.99 | 0.9, 0.99 | 0.5, 0.9 | same |
| Coupling block | cubic spline | same | same | same |
| # spline bins | 60 | same | same | same |
| # coupling blocks | 25 | 25 | 25 | 20 |
| Layers per block | 3 | 3 | 3 | 6 |
| # generated events | 2M | 2M | 2M | 1M |

servables of the base networks or the lower-multiplicity network, and on the number of jets. Because the $\mu\mu j$ and $\mu\mu jj$ networks are trained on events with mixed multiplicities, we guarantee a balanced training by drawing a random subset of the training data set at the beginning of each epoch containing equal numbers of events from all different multiplicities. While all three networks are trained separately, they are combined as a generator. We have found this conditional network architecture to provide the best balance of training time and performance.

Our network is implemented using PYTORCH [64] with the ADAM optimizer [65], and a one-cycle learning-rate scheduler [66]. The affine coupling blocks of the standard conditional INN setup [51,67] are replaced by cubic spline coupling blocks [68], which are more efficient in learning complex phase space patterns precisely and reliably. The coupling block splits the target space into bins of variable width based on trainable support points, which are connected with a cubic function. They are combined with random but fixed rotations to ensure interaction between all input variables. The parameter ranges of input, output and intermediate spaces are limited to $[-10, 10]$ on both sides of the coupling blocks, numbers outside this range are mapped onto themselves. The individual coupling blocks split their input vector in two halves $(u_i, v_i)$ and transforms $v_i$ as

$$v_i{}' = s(v_i; \chi(u_i, c_i)).$$ (5)

The $c_i$ are the conditional inputs of the network. The function $\chi$ is a fully connected sub-network with $2n_{\text{bins}} + 2$ outputs, where $n_{\text{bins}}$ is the number of spline bins. They encode the horizontal and vertical positions of the spline knots and its slope at the boundaries. The loss function for a cINN can most easily be defined in terms of the ratio of the intractable reference density $P_{\text{data}}(x; c)$ and the learned or model density $P(x; c)$ in terms of the phase space position $x$ and the condition $c$. We can ignore the normalization $\log P_{\text{data}}(x; c)$, because it does not affect the network training,

$$\begin{aligned} \mathcal{L}_G &= -\int dx \, P_{\text{data}}(x, c) \, \log \frac{P(x; c)}{P_{\text{data}}(x; c)} \\ &= -\int dx \, P_{\text{data}}(x, c) \, \log P(x; c) + \text{const} \\ &= -\int dx \, P_{\text{data}}(x, c) \left[ \log P_{\text{latent}}(\psi(x; c)) + \log J(x; c) \right] + \text{const}. \end{aligned}$$ (6)

In the last line we change variables between phase space and latent space and split $P(x; c)$ into an the latent-space distribution in terms of the INN-encoded mapping $\psi$ and its Jacobian $J$. Assuming a Gaussian in the latent space this gives us for a batch of $B$ inputs

$$\mathcal{L}_G \approx \sum_{i=1}^{B} \left( \frac{\psi(x_i; c_i)^2}{2} - \log J_i \right).$$ (7)

We list all hyperparameters in Tab. 1.

**Magic transformation**

A major challenge of the $Z+$ jets final state is illustrated in Fig. 2, where we show the $\Delta\phi$ vs $\Delta\eta$ correlations for the exclusive 2-jet sample. We see that most events prefer a back-to-back topology, but a small number of events features two jets recoiling against the $Z$, cut off by the requirement $\Delta R_{jj} > 0.4$. The ring around the corresponding small circle is a local maximum, and inside the ring the phase space density drops to zero. Because this entire structure lives in a poorly populated phase space region, the INN typically ignores the local maximum and smoothly interpolates over the entire ring-hole structure. We emphasize that in our case this

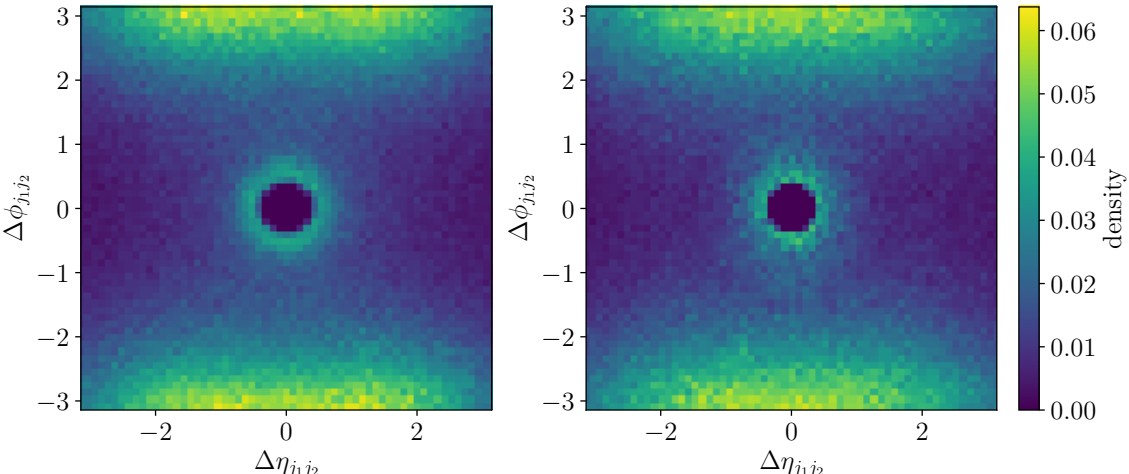

Figure 2: Jet-jet correlations for events with two jets. We show truth (left) and INN-generated events (right).

problem is not caused by the non-trivial phase space topology [69], the network interpolates smoothly through the holes, but a problem of the precision with which the network learns features just around these holes.

We can improve our network performance, after noticing the issue, by using some physics intuition and observing a near-magic aspect of network training. To this end, we map out the local maximum structure and make use of the fact that our network is extremely efficient at interpolating smooth functions. To exploit this property we define a $\Delta R_{jj}$-dependent transformation which turns the actual phase space pattern into a smoothly dropping curve, let the network learn this smooth function extremely well, and then undo the transformation to re-build the local maximum pattern. A simple smoothing function for our case is

$$
f(\Delta R) = \begin{cases} 0, & \text{for } \Delta R < R_-, \\ \dfrac{\Delta R - R_-}{R_+ - R_-}, & \text{for } \Delta R \in [R_-, R_+], \\ 1, & \text{for } \Delta R > R_+. \end{cases} \tag{8}
$$

The transition region is defined such that it includes the cutoff to ensure non-vanishing weights, $R_- < R_{\min} = 0.4$, and its upper boundary is in a stable phase space regime. In our case we use $R_- = 0.2$ and $R_+ = 1.5$ without much fine-tuning. We also apply this transformation to the 3-jet sample, where all $\Delta R_{jj}$-distribution have similar challenges, through additional event weights

$$
\begin{aligned}
w^{(\text{1-jet})} &= 1, \\
w^{(\text{2-jet})} &= f(\Delta R_{j_1,j_2}), \\
w^{(\text{3-jet})} &= f(\Delta R_{j_1,j_2}) f(\Delta R_{j_2,j_3}) f(\Delta R_{j_1,j_3}).
\end{aligned} \tag{9}
$$

After training the INN generator on these modified events we also enforce the jet separation and set all event weights with $\Delta R_{jj} < \Delta R_{\min}$ to zero. The inverse factor compensating for our magic transformation is then

$$
\tilde{f}(\Delta R) = \begin{cases} 0, & \text{for } \Delta R < R_{\min}, \\ \dfrac{R_+ - R_-}{\Delta R - R_-}, & \text{for } \Delta R \in [R_{\min}, R_+], \\ 1, & \text{for } \Delta R > R_+. \end{cases} \tag{10}
$$

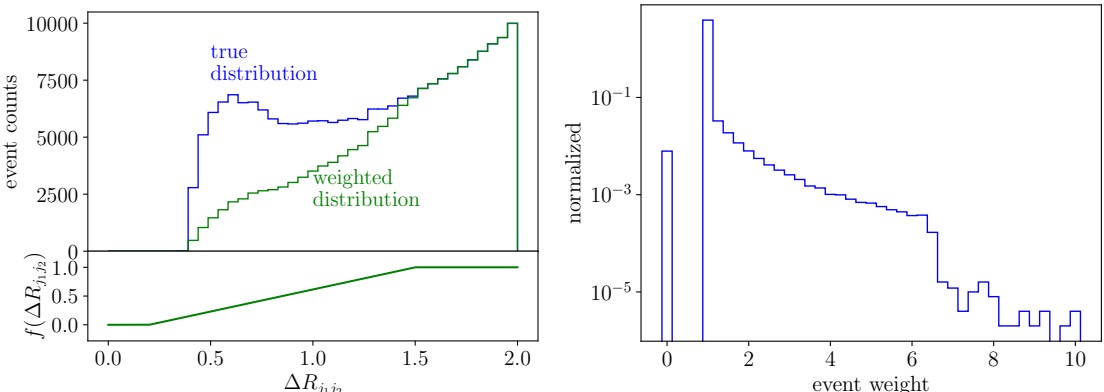

Figure 3: Left: $\Delta R_{j_1 j_2}$-distribution for $Z + 2$ jets events before and after the transformation of Eq.(9). Right: histogram of the weights of the generated events.

To train the INN generator on weighted data the loss function of Eq.(7) has to be changed to

$$\mathcal{L}_G = \sum_{i=1}^{B} \left( \frac{\psi(x_i; c_i)}{2} - J(x_i) \right) \frac{w(x_i)}{\sum_{i=1}^{B} w(x_i)} , \tag{11}$$

per batch with size $B$. Here, the weights are defined in Eq.(9), $x_i$ are the latent space vectors, and $J_i$ are the corresponding logarithms of the Jacobian. In the right panel of Fig. 2 we see that our network architecture indeed captures the intricate structure in the jet-jet correlations. The distribution of the resulting event weights is shown in Fig. 3. By construction all finite event weights are above one, and hardly any of them reach values for more than seven, which implies that these weights can be easily removed by standard reweighting techniques.

Our magic transformation is similar to a refinement, defined as per-event modifications of phase space distributions [70], whereas reweighting uses weights for individual phase space points or events to improve the agreement between generator output and truth [71]. However, our transformation is, by standard phase-space mapping arguments, counter-intuitive.[1] Instead of removing a leading dependence from a curve and learning a small but non-trivial difference, we smooth out a subtle pattern and rely on an outstanding network interpolation to learn the smoothed-out function better than the original pattern. This is motivated by the way flow networks learn distributions, which is more similar to a fit than a combination of local patterns [59]. The technical disadvantage of the smoothing transformation is that the generated events are now weighted, its advantage is that it is very versatile. Another disadvantage is that it needs to be applied based on an observed deficiency of the network and does not systematically improve the training of generative INNs, so below we will try to find alternative solutions to improve the network performance.

**INN-generator benchmark**

In Fig. 4 we show a set of kinematic distributions for our training data, truth defined as a statistically independent version of the training sample, and the output of the INN-generator with the magic transformation of Eq.(9). We show distributions for exclusive $Z + \{1, 2, 3\}$ jets samples and define the relative deviation for binned kinematic distributions as

$$\delta[\%] = 100 \frac{|\text{Model} - \text{Truth}|}{\text{Truth}} . \tag{12}$$

---

[1]As a matter of fact, our magic transformation of the density is the exact opposite of the standard phase space mapping for Monte Carlo integration.

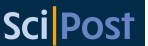

Figure 4: INN distributions for $Z+1$ jet (upper), $Z+2$ jets (middle), $Z+3$ jets (lower left) and an inclusive distribution (lower right) from a combined $Z+$ jets generation. We show weighted events using the magic transformation of Eq.(9) to improve the $\Delta R$ distributions.

In the top row the final state consists of the $Z$-decay products and one recoil jet, and we see that the recoil spectrum as well as the sharp $Z$-mass are learned with high precision. That remains true when we add a second jet, including the critical $\Delta R_{j_1 j_2}$ correlation discussed above. Finally, adding yet another jet the network learns the complete set of angular jet-jet correlations. Looking a the precision of the training sample, which consists of half of our full data set, we see that at least in the bulk of the kinematic distribution, the training data set agrees with truth at the percent level or better. This changes in the kinematic tails, where the statistical precision of the training data drops continuously. The level of agreement between the INN-generated events and truth also reaches the percent level in densely populated phase space regions, but it is slightly worse than the precision of the training sample. Also the $Z$-

peak even in the 1-jet sample is not perfectly learned by the INN, which leaves us a little bit of work to do on the precision side. To confirm that the differemt jet-exclusive samples are combined correctly, we show the hadronic $H_T$ or scalar sum of all transverse jet momenta in the lower-right panel. Its precision nicely tracks that of the different $p_{T,j}$ distributions.

## 3 DiscFlow generator

One way to systemically improve and control a precision INN-generator is to combine it with a discriminator. It is inspired by incredibly successful GAN applications also in LHC simulations [2, 3, 72]. In our preliminary studies we never reached a sufficient precision with established GAN architectures [36], while INN-generators proved very promising [59]. Compared to reweighting and refinement methods, a GAN-like setup has the advantage that the generator and discriminator networks already communicate during the joint training. We will show how such a discriminator network can be used to improve precision event generation and then show how a discriminator can be coupled to our INN generator in a new DiscFlow architecture.

### 3.1 Discriminator reweighting

Before we train our INN-generator jointly with a discriminator, we illustrate the power of such a discriminator by training it independently and reweighting events with the discriminator output [71]. This requires that our discriminator output can eventually be transformed into a probabilistic correction. We train a simple network described in Tab. 2 by minimizing the cross entropy to extract a probability $D(x_i) \to 0(1)$ for an identified generator (truth) event $x_i$. For a perfect generated sample the discriminator cannot tell generated events from true events, and the output becomes $D(x_i) = 0.5$ everywhere. Using this discriminator output we define the event weight

$$w_D(x_i) = \frac{D(x_i)}{1 - D(x_i)} \to \frac{P_{\text{data}}(x_i)}{P(x_i)}. \tag{13}$$

In the conventions of Eq.(6) $w_D$ approximates the ratio of true over generated phase space densities, so we can use it to reweight each event such that it reproduces the true kinematic distributions at the level they are encoded in the discriminator.

To see how precisely this kind of discriminator works we use the standard INN generator from Sec. 2.2. We omit the magic transformation described in Eq.(9), to define a challenge for

Table 2: Training setup and hyperparameters for the discriminator.

| hyper-parameter | value |
| --- | --- |
| LR scheduling | Reduce-on-plateau |
| Starter LR | $1 \cdot 10^{-2}$ |
| Epochs | 200 |
| Batch size | 2048 |
| Adam $\beta_1, \beta_2$ | 0.5, 0.9 |
| Layer type | Dense |
| Number layers | 8 |
| Internal size | 256 |



Figure 5: Discriminator-reweighted INN distributions for $Z+1$ jet (upper), $Z+2$ jets (middle), and $Z+3$ jets (lower) from a combined $Z+$ jets generation. The bottom panels show the average correction factor obtained from the discriminator output, the INN results without reweighting are the same as in Fig. 4, except for slightly longer training.

the discriminator. For each jet-multiplicity of the cINN model, we train a discriminative model in parallel to the generative model, but for now without the two networks communicating with each other. The input to the three distinct discriminator networks, one per multiplicity, are the usual observables $p_T, \eta, \phi$, and $m$ of Eq.(3) for each final-state particle. We explicitly include a set of correlations known to challenge our naive INN generator and train the discriminator

$$\mathcal{L}_D = -\sum_i^B \log(1 - D(x_{i,\text{gen}})) - \sum_i^B \log(D(x_{i,\text{data}})), \qquad (14)$$

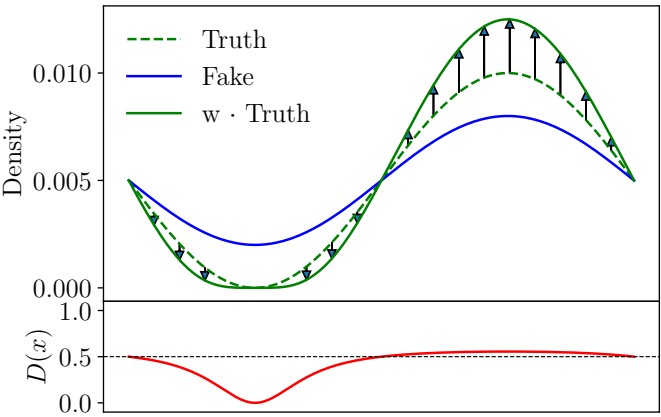

Figure 6: Illustration of the DiscFlow method. Weights computed by the discriminator shift the reference (true) density downwards whenever the generator (fake) distribution overshoots and vice-versa. This way the deviations of the to-be-trained generator density are over-exaggerated.

with generated vectors extended depending on the jet multiplicity

$$x_i = \{p_{T,j}, \eta_j, \phi_j, M_j\} \cup \{M_{\mu\mu}\} \cup \{\Delta R_{2,3}\} \cup \{\Delta R_{2,4}, \Delta R_{3,4}\},\tag{15}$$

and corresponding training vectors $x_{i,\text{data}}$.

In Fig. 5 we show sample kinematic distributions for the $Z + \{1, 2, 3\}$ jet final states. Truth is defined as the high-statistics limit of the training data. The INN events are generated with the default generator, without the magic transformation of Eq.(9), so they are unweighted events. The reweighted events are post-processed INN events with the average weight per bin shown in the second panel. While for some of the shown distribution a flat dependence $w_D = 1$ indicates that the generator has learned to reproduce the training data to the best knowledge of the discriminator, our more challenging distributions are significantly improved by the discriminator. That includes the reconstructed $Z$-mass as well as the different $\Delta R_{jj}$-distributions.

Comparing the discriminator-reweighted performance to the magic transformation results in Fig. 4, reproduced as the blue lines in Fig. 5, we see that the tricky distributions like $\Delta R_{j_1 j_2}$ or $\Delta R_{j_1 j_3}$ are further improved through the reweighting over their entire range. For the comparably flat $p_T$-distributions the precision of the reweighted events is becoming comparable to the training statistics, both for the bulk of the distribution and for the sparsely populated tails. Of all kinematic distributions we checked, the vector sum of all hard transverse momenta of the 5-object final state is the only distribution where the naive INN-generator only learns the phase space distribution only at the 10% level. Also those are corrected fine by the discriminator reweighting.

While the discriminator reweighting provides us with an architecture that learns complex LHC events at the percent level or at the level of the training statistics, it comes with the disadvantage of generating weighted events and does not use the opportunity for the generator and discriminator to improve each other. Both of these open questions will be discussed in the next architecture.

## 3.2 Joint training

After observing the benefits from an additional discriminator network, the question is how we can make use of this second network most efficiently. If it is possible to train the discriminator

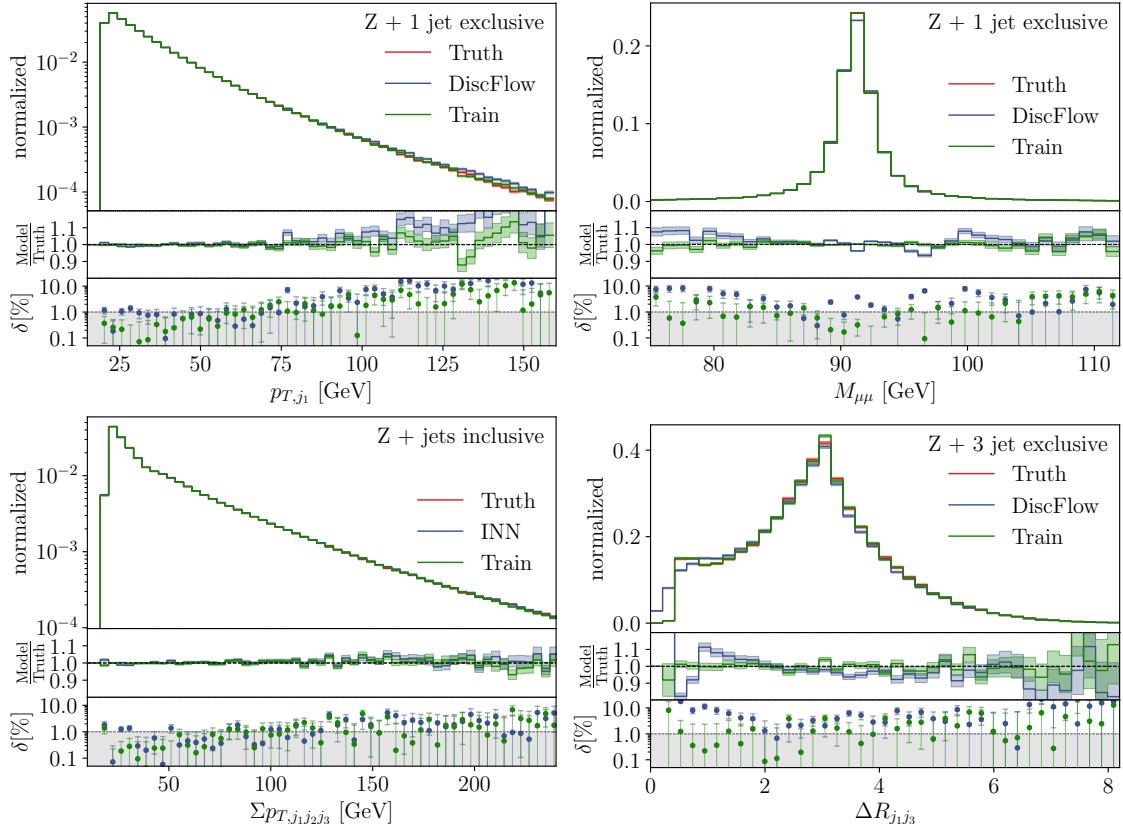

Figure 7: DiscFlow distributions for $Z+1$ jet, $Z+3$ jets, and an inclusive distribution from a combined $Z+$ jets generation after joint generator–discriminator training.

and generator network in parallel and give them access to each other, a joint GAN-like setup could be very efficient [73]. Unfortunately, we have not been able to reach the required Nash equilibrium in an adversarial training for our specific INN setup. Instead, one of the two players was always able to overpower the other.

Instead of relying on a Nash equilibrium between the two competing network architectures we can avoid a two-part loss functions entirely and incorporate the discriminator information into the generator loss of Eq.(7) through the event weight function $w_D(x)$ of Eq.(13),

$$
\begin{aligned}
\mathcal{L}_{\text{DiscFlow}} &= -\sum_{i=1}^{B} w_D(x_i)^\alpha \, \log \frac{P(x_i; c_i)}{P_{\text{data}}(x_i; c_i)} \\
&\approx -\int dx \, \frac{P_{\text{data}}^{\alpha+1}(x)}{P^\alpha(x)} \, \log \frac{P(x)}{P_{\text{data}}(x)} \\
&= -\int dx \left( \frac{P_{\text{data}}(x)}{P(x)} \right)^{\alpha+1} P(x) \log P(x) + \int dx \left( \frac{P_{\text{data}}(x)}{P(x)} \right)^{\alpha} P_{\text{data}}(x) \log P_{\text{data}}(x) \\
&= -\left\langle \left( \frac{P_{\text{data}}(x)}{P(x)} \right)^{\alpha+1} \log P(x) \right\rangle_P + \left\langle \left( \frac{P_{\text{data}}(x)}{P(x)} \right)^{\alpha} \log P_{\text{data}}(x) \right\rangle_{P_{\text{data}}},
\end{aligned}
\tag{16}
$$

with an appropriately defined expectation value. For the continuum limit we omit the conditional argument and assume a perfectly trained discriminator. Note that in our simple DiscFlow setup the discriminator weights $\omega_D \approx P_{\text{data}}(x)/P(x)$ do not have gradients with respect to the generative model parameters, so only the first term in the last line contributes to the optimization. This term corresponds to the negative log-likelihood of training samples drawn from the

weighted truth distribution. The hyperparameter $\alpha$ determines the impact of the discriminator output, and we introduce an additional discriminator dependence as

$$\alpha = \alpha_0 \left| \frac{1}{2} - D(x) \right| . \tag{17}$$

During training we increase $\alpha_0$ linearly to enhance the impact of the reweighting factor, while the improved training will drive the discriminator to $D(x) \to 1/2$. This functional form for $\alpha$ is the simplest way of combining the two effects towards a stable result.

From Eq.(16) we see that our modified loss is equivalent to training on a shifted reference distribution. In Fig. 6 we illustrate what happens if the generator populates a phase space region too densely and we reduce the weight of the training events there. Conversely, if a region is too sparsely populated by the generator, increased loss weights amplify the effect of the training events. Our new discriminator–generator coupling through weights has the advantage that it does not require a Nash equilibrium between two competing networks, so the discriminator can no longer overpower the generator. As the generator converges towards the true distribution, the discriminator will stabilize as $w_D(x) \to 1$, and the generator loss will approach its unweighted global minimum.

When training the two DiscFlow networks jointly, we split the batches per epoch equally between both networks, training each network on a separate subset of the training data. To increase the stability, we start by training the generator and the separate discriminators for the different jet multiplicities separately and only combine them to a stable joint training once all networks are pre-trained.

In Fig. 7 we show the performance of the DiscFlow setup to our $Z$+jets benchmark process. First, we see that in the bulk of the flat distributions like $p_{T,j}$ the generator reproduces the correct phase space density almost at the level of the training statistics. Comparing the results to Fig. 4 and Fig. 5 we see a comparable, possibly improved, performance of the joint training. The non-negligible density of generated events below the cut at $\Delta R = 0.4$ shows that the DiscFlow method is only effective in phase space regions populated by training data. These results indicate that the joint training of the generator with a discriminator corrects the invariant mass and all other tricky distributions almost to the level of the training statistics, but with unweighted events, unlike for the magic transformation in Fig. 4 and the explicit reweighting in Fig. 5.

In the ideal AI-world we assume that after successful joint training the discriminator will have transferred all of its information into the generator, such that $D(x) = 0.5$ at any point of phase space. In reality, this is not at all guaranteed. We know from Fig. 5 that the discriminator can learn the $\Delta R$ features very well, so we combine the joint training and discriminator reweighting strategies to ensure that we extract the full performance of both networks. In Fig. 8 we show the same training results as in Fig. 7, but reweighted with $w_D$. We see that the reweighting leads to a small correction of the $M_{\mu\mu}$-distribution and a sizeable correction to the $\Delta R_{jj}$ features close to the jet separation cut. Because of the way we provide the event input, we note that the transverse momentum conservation would become the next challenge after mastering $M_{\mu\mu}$ and $\Delta R_{jj}$. For all other observables our reweighted DiscFlow network indeed reproduces the true kinematic distributions at the percent level provided by the training statistics.

While in Fig. 8 we see that the correction factor obtained from the discriminator shows the agreement of training events and simulated events, it is crucial that we search the fully exclusive phase space for systematic deviations between training and simulated events. In Fig. 9 we histogram all event weights $w_D(x_i)$ for $Z+$ jets production. For the high-statistics $Z + 1$ jet sample the correction weights are at most at the percent level. The fact that our generator only learns the phase space density and not the total rates allows for a slight bias



Figure 8: Discriminator-reweighted DiscFlow distributions for $Z + 1$ jet (upper), $Z + 2$ jets (middle), and $Z + 3$ jets (lower) from a combined $Z+$ jets generation. The bottom panels show the average correction factor obtained from the discriminator output. The DiscFlow results for joint generator–discriminator training without reweighting are the same as in Fig. 7.

in the event weight distributions. For the bulk of the kinematic distributions the bin-wise correction in Fig. 8 is still slightly smaller than the weights shown here, which means that some of the corrections are simply noise. The width of the weight distribution increases for higher jet multiplicities, simply reflecting the drop in training statistics. Combining Fig. 9 and Fig. 8 allows us to trace the large weights $w_D$ to critical phase space regions, like the lower tail of the $M_{\mu\mu}$-distribution for $Z + 1$ jet or $\Delta R_{jj} \lesssim 0.5$ for $Z + 2/3$ jets.

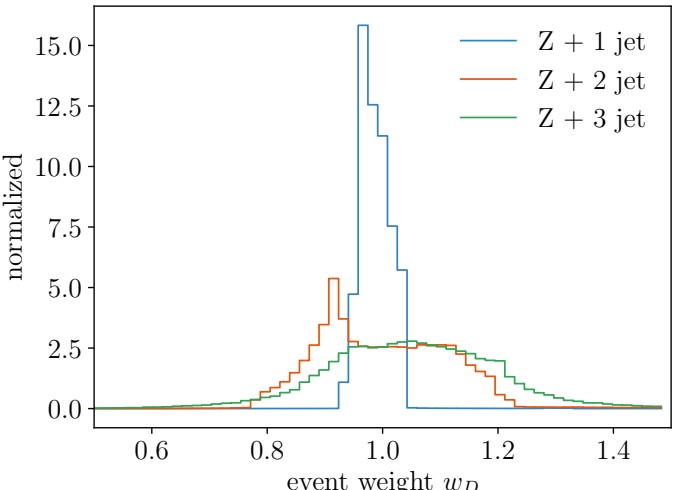

Figure 9: Distribution over the weights $w_D$ computed over the entire, not marginalized phase space.

## 4 Uncertainties and control

After introducing our precision generator architecture in Sec. 2 and extending it to a discriminator-generator architecture for control in Sec. 3, the last item on our list of LHC tasks is a comprehensive treatment of uncertainties. A proper uncertainty treatment has been discussed for instance for regression or classification networks [57, 58, 74], while for generative networks there exists only a first study on how to use and interpret Bayesian INNs [59]. In this final section we discuss how different uncertainties on generated events can be extracted using a Bayesian generator network, a conditional sampling using simulated uncertainties, and the discriminator introduced in the previous section. Each of these handles allows us to control certain kinds of uncertainties, and in combination they allow us to extract a meaningful uncertainty map over phase space.

### 4.1 Bayesian network

The simple idea behind Bayesian networks is to replace trained network weights by trained distributions of network weights. If we evaluate the network by sampling over these distributions, the network output will be a central value of the numerically defined function and an uncertainty distribution [53–55]. Because general MCMC-methods become expensive for larger networks, we rely on variational inference to generate the weight distributions [75]. More specifically, we rely on a Gaussian approximation for the network weight distribution and learn the mean and the standard deviation instead of just one value in a deterministic network. Because of the non-linear nature of the network the output does not have a Gaussian uncertainty distribution [58]. Our Bayesian INN (BINN) follows the same setup as our deterministic INN-generator in Sec. 2.2, converted to the Bayesian setup following Ref. [59].

For a Bayesian generative network we supplement the phase space density $p(x)$, encoded in the density of unweighted events, with an uncertainty map $\sigma(x)$ over the same phase space. To extract the density we bin events in a histogram for a given observable and with finite statistics. Focussing on one histogram and omitting the corresponding phase space argument $x$ the expected number of events per bin is

$$\mu \equiv \langle n \rangle = \sum_n n P_N(n), \tag{18}$$

with $P_N(n)$ given by the binomial or Poisson probability of observing $n$ events in this bin. This event count should be the mean of the BINN distribution, defined by sampling from the distribution $q(\theta)$ over the network weights $\theta$,

$$\langle n \rangle = \int d\theta \, q(\theta) \sum_n n P_N(n|\theta) \equiv \int d\theta \, q(\theta) \, \langle n \rangle_\theta \,. \tag{19}$$

Following the same argument as in Ref. [58] we can compute the standard deviation of this sampled event count and split it into two terms,

$$\begin{aligned}
\sigma_{\text{tot}}^2 &= \langle (n - \langle n \rangle)^2 \rangle \\
&= \int d\theta \, q(\theta) \left[ \langle n^2 \rangle_\theta - 2\langle n \rangle_\theta \langle n \rangle + \langle n \rangle^2 \right] \\
&= \int d\theta \, q(\theta) \left[ \langle n^2 \rangle_\theta - \langle n \rangle_\theta^2 + (\langle n \rangle_\theta - \langle n \rangle)^2 \right] \equiv \sigma_{\text{stoch}}^2 + \sigma_{\text{pred}}^2 \,.
\end{aligned} \tag{20}$$

The first contribution to the uncertainty is the variance of the Poisson distribution,

$$\sigma_{\text{stoch}}^2 = \int d\theta \, q(\theta) \left[ \langle n^2 \rangle_\theta - \langle n \rangle_\theta^2 \right] = \langle n \rangle \,. \tag{21}$$

Even if the network is perfectly trained and $q(\theta)$ turns into a delta distribution, it does not vanish, because it describes the stochastic nature of our binned data set. The second term,

$$\sigma_{\text{pred}}^2 = \int d\theta \, q(\theta) \left[ \langle n \rangle_\theta - \langle n \rangle \right]^2 \,, \tag{22}$$

captures the deviation of our network from a perfectly trained network, where the widths of the network weights vanish.

Moving from a binned to a continuous distribution we can transform our results into the density and uncertainty maps over phase space, as introduced in Ref. [59]. Assuming $\langle n \rangle \propto p(x)$, with an appropriate proportionality factor and a continuous phase space variable $x$, Eqs.(19) and (22) turn into

$$\begin{aligned}
p(x) &= \int d\theta \, q(\theta) \, p(x|\theta), \\
\sigma_{\text{pred}}^2(x) &= \int d\theta \, q(\theta) \, \left[ p(x|\theta) - p(x) \right]^2 \,.
\end{aligned} \tag{23}$$

To estimate $\sigma_{\text{tot}}$, we sample $\theta$ and $n$ from their underlying distributions and compute $\langle n \rangle$. In practice, we draw weights $\theta$, generate $N$ events with those weights, histogram them for the observable of interest, extract $n$ per bin. Because the INN-generator is very fast, we can repeat this process to compute the standard deviation. To see the effect of the different contributions to the BINN uncertainty we illustrate the correlation between the event count and $\sigma_{\text{tot}}$ for $Z + 1$ jet events in Figure 10, with the $p_{T,j}$-distribution described by 60 bins. Each of these bins corresponds to a dot in the figure. As long as our sampling is limited by the statistics of the generated events we find the expected Poisson scaling $\sigma \propto \sqrt{\mu}$, corresponding to the contribution $\sigma_{\text{stoch}}$. For larger statistics, $\sigma_{\text{stoch}}$ becomes relatively less important, and the actual predictive uncertainty of the BINN takes over, $\sigma_{\text{tot}} \approx \sigma_{\text{pred}}$.

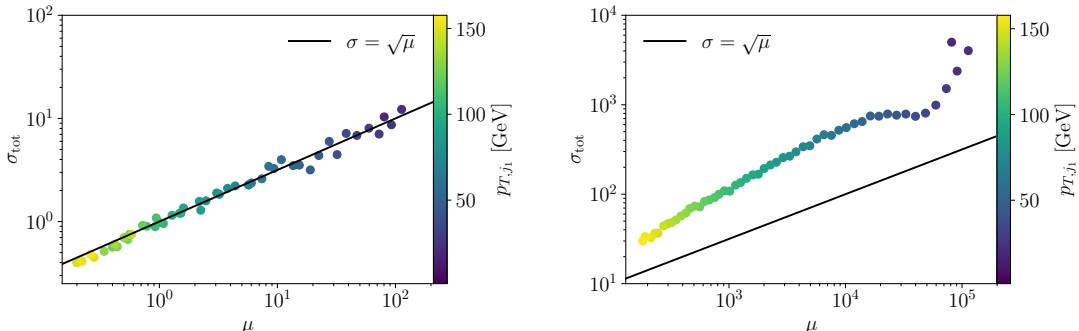

Figure 10: Correlation between event count and BINN uncertainty for 1000 (left) and 1M (right) generated events. The diagonal like defines the Gaussian scaling for a statistically limited sample.

**Sources of uncertainties**

By construction, Bayesian networks capture the effects of limited training statistics and non-perfect training. If we control the truth information and can augment the training data, a Bayesian network can also propagate the effects of systematic biases, systematic uncertainties, or noise into the network output [57, 58]. For generative networks, the Bayesian network is ideally suited to understand the way the network learns the phase space density by following the density map it learns in parallel [59]. As a side remark, we can use this information to track the learning of the BINN for our $Z$+jets events. We find that the network first learns the $p_T$-distributions of the different final-state particles quite precisely, before it targets the angular correlations. This explains why small features of the $\Delta R$-distributions are the hardest to learn, because they arise only for the correlation of the $\Delta\eta$ and $\Delta\phi$ observables. Correspondingly, we find that one way of improving the performance on the angular correlation is to apply noise specifically to the $p_T$-distributions. On the other hand, the magic transformation of Eq.(9) turns out to be the more efficient solution to this problem, so we also apply it to the BINN.

When modelling different uncertainties, the problem with augmented training data for generative networks is that their training is, strictly speaking, unsupervised. We do not have access to the true density distribution and have to extract it by binning event samples. This means that additional noise will only be visible in the BINN uncertainty if it destabilizes the training altogether. Other data augmentation will simply lead to a different target density, overriding the density encoded in the original set of events. This is why in the following we will discuss training statistics and stability, and postpone the description of systematics in generative network training to Sec. 4.2.

In Fig. 11 we show the uncertainty $\sigma_{\text{tot}} \approx \sigma_{\text{pred}}$ given by the BINN for a Bayesian version of the network introduced in Sec. 2.2, including the magic transformation for the $\Delta R$-distributions. As before, we see that the network learns the phase space density very precisely for simple kinematic distributions like $p_{T,j_1}$. The slightly worse performance compared to the deterministic network in Fig. 11 is due to the increased training effort required by the larger network. The extracted uncertainties for $p_{T,j_1}$ and $p_{T,j_2}$ for instance in the bulk reflect the lower statistics of the $Z + 2$ jet training sample compared to $Z + 1$ jet. The narrow $M_{\mu\mu}$-distribution challenges the uncertainty estimate in that the network learns neither the density nor the uncertainty very precisely [59]. This limitation will be overcome once the network learns the feature in the density properly. For the different $\Delta R$-distributions we see that the network learns the density well, thanks to the magic transformation of Eq.(9). Therefore, the network also reports a comparably large uncertainty in the critical phase space regions around



Figure 11: BINN densities and uncertainties for $Z + 1$ jet (upper), $Z + 2$ jets (middle), and $Z + 3$ jets (lower) from a combined $Z+$ jets generation. The architecture and training data correspond to the deterministic network results shown in Fig. 4, including the magic transformation of Eq.(9).

$$\Delta R_{ij} = 0.4 \dots 1.$$

**Effect of training statistics**

From the above discussion it is clear that one way to test the BINN uncertainties is to train the same network the same way, but on training samples of different size. We start with one batch size, 3072 events, and increase the training sample to the maximum of 2.7M. For $Z + 1$ jet we show the relative uncertainty as a function of transverse momenta, for instance, in Fig. 12. In both cases we see that over most the distribution the uncertainty improves with the training statistics. However, we also see that in the right tail of the $p_{T,\mu_1}$ distribution the lowest-statistics trainings does not estimate the uncertainty correctly. Again, this reflects the fact that, if the

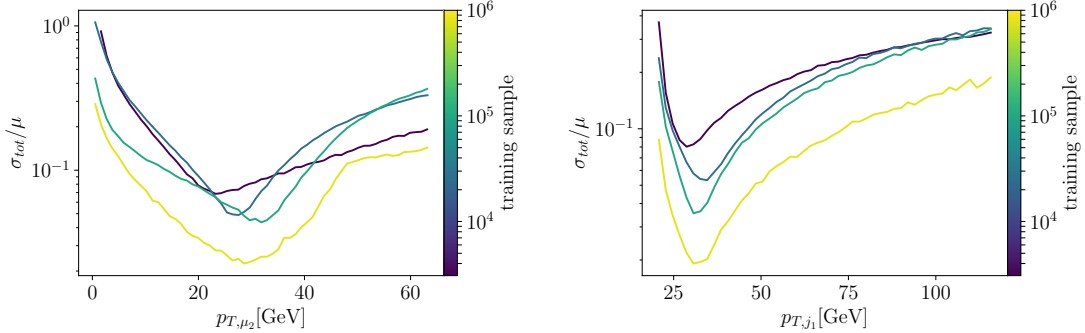

Figure 12: Relative uncertainty from the BINN for the $Z + 1$ jet sample, as a function of the size of the training sample.

network does not even have enough data to estimate the density, it will not provide a reliable uncertainty estimate. For $p_{T,j_1}$ this effect does not occur, even in the tails where the network has to extrapolate eventually.

## 4.2 Conditional augmentations

As discussed above, Bayesian generative networks will not capture typical systematic or theory uncertainties. Those uncertainties are known, for instance as limitations to predict or reconstruct objects in certain phase space regions, but unlike for regression or classification networks we cannot augment the training date to account for them. The reason is that generative networks extract the underlying phase space density implicitly, so we cannot control what augmented training data actually does to the network training.

For illustration purpose, let us introduce a toy theory uncertainty proportional to the transverse momentum of a jet. This could incorporate the limitation of an event generator, based on perturbative QCD, in predicting tails of kinematic distributions inducing large logarithms. In terms of a nuisance parameter $a$ such an uncertainty would shift the unit weights of our training events to

$$w = 1 + a \left( \frac{p_{T,j_1} - 15 \text{ GeV}}{100 \text{ GeV}} \right)^2 , \tag{24}$$

where the transverse momentum is given in GeV, we account for a threshold at 15 GeV, and we choose a quadratic scaling to enhance the effects of this shift in the tails.

Instead ot just augmenting the training data, we train the network conditionally on this nuisance parameter and then sample the nuisance parameter for the trained network, to reproduce the systematic or theory uncertainty now encoded in the network. This means we then our Bayesian INN conditionally on values $a = 0 \dots 30$ in steps of one. For the event generation incorporating the theory uncertainty we can sample kinematic distributions for different $a$-values. In Fig. 13 we show generated distributions for different values of $a$. To model the conditional parameter similar to phase space and allow for an uncertainty on the conditional nuisance parameter, we sample $a$ with a Gaussian around its central value and a standard deviation of $\min(a/10, 0.1)$. The two panels show the modified $p_{T,j_1}$-distribution and its impact on $p_{T,j_2}$ through correlations. As expected, the effects are similar, but the multiparticle recoil washes out the effects on $p_{T,j_2}$. In the upper panels we compare the effect of the theory uncertainty $a = 0 \dots 12$ to the statistical training uncertainty given by the BINN. We see that our method traces the additional theory or systematic uncertainty, and allows us

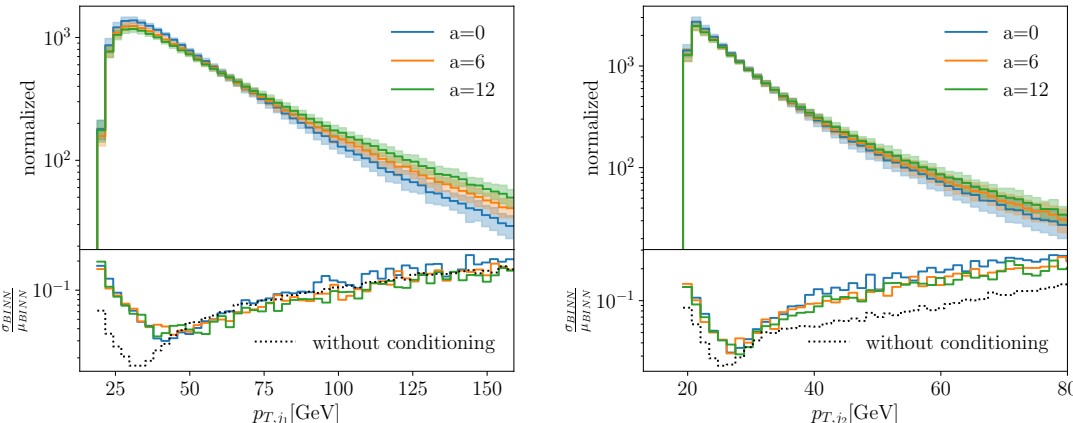

Figure 13: BINN densities for $Z+$ jets and conditional training with an enhanced-tail augmentation in $p_{T,_1}$, as defined in Eq.(24).

to reliably estimate its sub-leading nature for $p_{T,j_2}$. While we show ranges of $a$, corresponding to the typical flat likelihood used for theory uncertainties, we could obviously sample the different $a$-values during event generation. In the lower panels we show the relative BINN uncertainties, to ensure that the training for the different $a$-values is stable. For $p_{T,j_1}$ the data augmentation has a slight chilling effect on the high-precision training around the maximum of the distribution. In the statistically limited tails towards larger $p_T$ the BINN training without and with augmentations behaves the same. Looking at the recoil correlation, the BINN reports a slightly larger uncertainty for the augmented training, correctly reflecting the fact that the network now has to learn an additional source of correlations. At least for the range of shown $a$-values this BINN uncertainty is independent of the size of the augmentation.

## 4.3 Discriminator for consistency

After introducing two ways of tracing specific uncertainties for generative networks and controlling their precision, we come back to the joint DiscFlow generator–discriminator training. In complete analogy to, for instance, higher-order perturbative corrections, we can use the jointly trained discriminator to improve the network precision and at the same time guide us to significant differences between training data and generated data. Because the discriminator is a simpler network than the INN-generator, it is well suited to search for deviations which the BINN misses in its density and uncertainty maps.

In Fig. 14 we illustrate the different aspects of our uncertainty-controlled precision generator. First, we see that the INN generator indeed learns and reproduces the phase space density at the level of the training statistics. In the remaining panels we show three ways to control possible uncertainty, using the discriminator, a BINN, and a BINN combined with augmented training data.

Each aspect is described in detail in this paper:

· joint discriminator–generator training (DiscFlow) for precision generation — Fig. 7;

· discriminator to control inconsistencies between training and generated events — Fig. 8;

· BINN to track uncertainty on the learned phase space density — Fig. 11;

· conditional augmentation for systematic or theory uncertainties — Fig. 13.

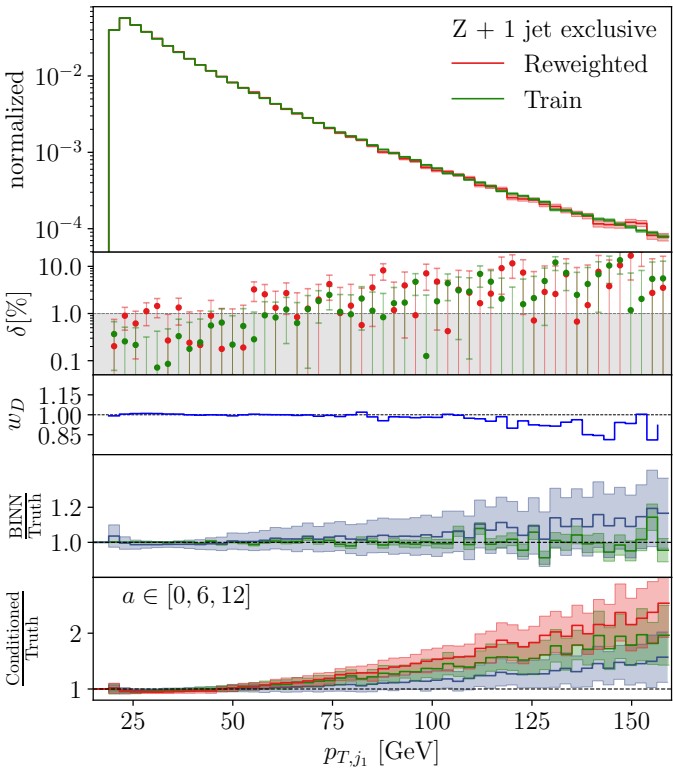

Figure 14: Illustration of uncertainty-controlled DiscFlow simulation. We show the reweighted $p_{T,j_1}$-distribution for the inclusive $Z$+jets sample, combined with the discriminator $D$, the BINN uncertainty, and the sampled systematic uncertainty defined through the data augmentation of Eq.(24).

## 5   Outlook

A crucial step in establishing generative networks as event generation tools for the LHC is the required precision in estimating the phase space density and full control of uncertainties in generated samples.

In the first part of this paper, we have shown how INN-generators can be trained on $Z$+jets events with a variable number of particles in the final state, to reproduce the true phase space density at the percent level, almost on par with the statistical uncertainty of the training sample. If we are willing to work with weighted events, with event weights of order one, we can either use a magic variable transformation or an additional discriminator network to achieve high precision all over phase space. Alternatively, we can train the discriminator jointly with the generator and use our novel DiscFlow architecture to provide unweighted events with high precision (Fig. 7). This joint training does not involve a Nash equilibrium and is especially stable. Any information that the discriminator has not transferred to the generator training can eventually be included through reweighting, giving our NN-event generator high precision combined with a high level of control (Fig. 8).

In the second part of this paper we have established three methods to control the precision INN-generator and its uncertainties. First, for unsupervised generative training we can use a Bayesian INN to estimate uncertainties from limited training statistics or sub-optimal network training (Fig. 11). Second, we can augment the training data conditionally on a nuisance parameter and sample this parameter to account for systematic or theory uncertainties including the full phase space correlations (Fig. 13). A reliable estimate of the different uncertainties allows us to compare the numerical impact of the different uncertainties. Finally,

we can use the jointly trained discriminator to identify phase space regions where the BINN lacks the necessary precision in its density and uncertainty maps over phase space.

All these aspects of our uncertainty-controlled precision generator are illustrated in Fig. 14. With this level of precision and control, INN-generators should be ready to be used as extremely efficient tools to generate LHC events. More generally, our study shows that generative INNs working on reconstructed objects can be used as reliable precision tools for a range of forward and inverse inference approaches as well as dedicated detector simulations.

## Acknowledgments

We would like to thank Ben Nachman and Jan Pawlowski for very helpful discussions on the DiskFlow loss function. In addition, we would like to thank Michel Luchmann and Manuel Haußmann for help with Bayesian networks and Luca Mantani and Ramon Winterhalder for their work on an earlier incarnation of this project. We are also very grateful to Ulli Köthe and Lynton Ardizzone for their expert advice on many aspects of this paper. The research of AB and TP is supported by the Deutsche Forschungsgemeinschaft (DFG, German Research Foundation) under grant 396021762 - TRR 257 Particle Physics Phenomenology after the Higgs Discovery. TH is supported by the DFG Research Training Group GK-1940, Particle Physics Beyond the Standard Model. The authors acknowledge support by the state of Baden-Württemberg through bwHPC and the German Research Foundation (DFG) through grant no INST 39/963-1 FUGG (bwForCluster NEMO). This work was supported by the Deutsche Forschungsgemeinschaft (DFG, German Research Foundation) under Germany's Excellence Strategy EXC 2181/1 - 390900948 (the Heidelberg STRUCTURES Excellence Cluster).

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
