# Peer review of "Generative Networks for Precision Enthusiasts"

_SciPost Physics, doi:SciPost Phys. 14, 078 (2023)_

## Round 2 · Referee Report · Anonymous · 2022-3-1

Strengths
The authors present a sophisticated generative model for LHC event generation capable of reproducing given training data to within few percent uncertainty. This is achieved by simultaneous training of an INN-type network and an additional discriminator that determines weights for correcting deviations between generative network and the training data. Employing Bayesian networks the authors manage to also account for uncertainties from the training as well as (in principle) systematic uncertainties associated with the target, e.g. from missing higher order perturbative corrections.
The presented methods are novel and innovative and the technical work is quite clearly described.
Weaknesses
While the reproduction of the input training data is quite impressive, the authors - to my taste - are not clear enough about the limitations of their approach and the actual use case for their method. What they achieve is a clone of the training data that, however, before got projected onto jet objects with fixed parameters/algorithm, thereby significantly reducing the dimensionality of the problem. While it is still an interesting and challenging problem, its practical applicability and use case are not quite clear.
Report
1) The authors should more clearly state the envisioned application scenario for their proposed method, thereby addressing both strengths and limitations of their technique.
To give an example, the process of choice is quite inclusive dilepton production, simulated with Sherpa. However, the authors then only consider the projection on fixed R=0.4 and pT>20 GeV jets and in fact only on exclusive jet multiplicities ... an horrendous simplification of the initial task.
2) The authors claim to account for the right admixture of the different jet multiplicities, however, results are only presented for exclusive jet bins. It would be good to show observables that receive contributions from all channels, for example the HT distribution, i.e. the scalar sum of jet pT's for the inclusive sample.
3) At several places the authors mention that their INN-generator is very fast, it would be nice to have a statement about the overall resources needed to generate events in their setup, though the comparison to the original generator might be misleading, see point 1.
Minor corrections:
- p11: '5-jet' -> '5-particle/object'
- p17: 'correlation of the ... correlations' ???
-p20: I do not see the claimed analogy to HO corrections, that needs further explanation
Requested changes
see above
Author: Theo Heimel on 2022-12-20 [id 3161]
(in reply to Report 1 on 2022-03-01)
1) The authors should more clearly state the envisioned application scenario for their proposed method, thereby addressing both strengths and limitations of their technique. To give an example, the process of choice is quite inclusive dilepton production, simulated with Sherpa. However, the authors then only consider the projection on fixed R=0.4 and pT>20 GeV jets and in fact only on exclusive jet multiplicities ... an horrendous simplification of the initial task. -> We noticed that we did not show any inclusive distributions, because they are less challenging that the exclusive ones, but we now state that out generator is jet-inclusive. We are not sure what aspects of an R-cut and a pT-cut is oversimplifying? We would argue that our method will work on all kinds of processes at the reconstruction level. We modified the beginning of 2.1 accordingly.
2) The authors claim to account for the right admixture of the different jet multiplicities, however, results are only presented for exclusive jet bins. It would be good to show observables that receive contributions from all channels, for example the HT distribution, i.e. the scalar sum of jet pT's for the inclusive sample. -> We added a comment on inclusive observables to the end of Sec.2 and included corresponding panels to Figs.4 and 7.
3) At several places the authors mention that their INN-generator is very fast, it would be nice to have a statement about the overall resources needed to generate events in their setup, though the comparison to the original generator might be misleading, see point 1. -> With our setup, generating 1M events takes roughly 2.5min on a GPU. However, because a comparison of the generation speed between different methods on different hardware might be misleading, and the focus of this work was exploring the toolbox provided by INNs for event generation, we did not include a benchmark of the generation time in the text.
Minor corrections:
- p11: '5-jet' -> '5-particle/object' -> done
- p17: 'correlation of the ... correlations' ??? -> done -p20: I do not see the claimed analogy to HO corrections, that needs further explanation -> We included a slightly longer discussion.
Author: Theo Heimel on 2022-12-20 [id 3160]
(in reply to Report 2 on 2022-05-01)Major:
ref' to
data' to make that clear. We keep it, because it is needed later.Minor:
Introduction

---

## Round 2 · Referee Report · Anonymous · 2022-5-1

(Invited Report)- Cite as: Anonymous, Report on arXiv:2110.13632v2, delivered 2022-05-01, doi: 10.21468/SciPost.Report.5008
Report
My sincere apologizes for the delayed report!
This paper presents and interesting study aimed at improving the precision of neural network surrogate models of event generators for collider physics. The paper is well written and contains a number of innovative studies. I am happy to recommend publication after the points below have been addressed.
Major:
- "To increase the efficiency, we stick to the same network for the common..." -> I found this paragraph to be quite confusing. Do you train a mumu+j network and then use that as input to the mumu + jj/jjj training? Aren't there correlations between the jets such that p(j1 j2) != p(j2|j1) x p(j1) ?
- What is P_{ref} in Eq. 6? it is never used again nor explained. Is it just to make the argument of the log dimensionless? Maybe it would be easier /clearer to drop it?
- What is \psi in Eq. 8? It is called "the network", but I thought you are predicting the full density P?
- "We emphasize that in our case this is not a problem with the non-trivial phase space topology" -> I did not understand this - isn't there actually a hole (in the topological sense) in the phase space?
- Performance: Figs. 4,5,7 don't look so different, although it is hard to compare them directly since they are spread across many pages. Can you comment on this?
- Eq. 18: This is mysterious to me - would you please provide some explanation? I understand that this downweights the w_D term when D is close to 1/2, as it should be if the learned and reference are similar. But why this functional form? Does adding the weight term and the D-dependence of alpha introduce a bias? Eq. 17 is no longer maximum likelihood so it is not obvious to me that its optimum is actuall the target density.
- The paragraph afer Fig. 7 makes it sound like the discriminator and generator networks are not trained at the same time. Why not?
- Fig. 10, right plot: Why does this also show sqrt(n) scaling?
- Fig. 12, left plot: Why does adding more data sometimes make the uncertainty worse?
Minor:
Introduction
- first paragraph: "analyses and inference" -> seems redundant; doing analysis is performing inference.
- Please spell out LHC the first time it is used. Same for GANs and VAEs in the second paragraph.
- second paragraph: "...potential to improve LHC simulations [1]" -> if I understand correctly, [1] is only about "physics generators", but does not include e.g. detector simulation (so I'm not sure it stands as a catch-all for this statement).
- second paragraph: seems odd that [33-37] and [38-48] are not in the previous paragraph, which is about the full range of steps in the simulation chain.
- third paragraph: GANs and VAEs are not invertible.
- Eq. 3: jets are not particles - perhaps "each physics object is..." would be more precise?
- "Magic Transformation" -> I find this a bit unsatisfying because this is something that is not automatically optimized, but manually put in by hand to fix an issue with the out of the box approach. What if your phase space is much bigger and there are more holes? Perhaps you could comment on how this generalizes? It may also be prudent to call it something other than "Magic" since that implies that it comes out of nowhere when in fact it is based on your physics intuition (!)
- Eq. 15 is confusing - y_i is usually a label, but I think here it has the same structure as Eq. 16? What is B? I would think the first sum would be the INN output and the second sum would be over the physics simulator output?
- Fig. 10/12: Why is the x-axis mu and not n (the symbol used for per-bin event counts in the text)

---

## Round 3 · List of Changes

See responses to the referees for a detailed list of changes.

---

## Editorial Decision

published